# Risk Classification of Bladder Cancer by Gene Expression and Molecular Subtype

**DOI:** 10.3390/cancers15072149

**Published:** 2023-04-04

**Authors:** Ana Blanca, Antonio Lopez-Beltran, Kevin Lopez-Porcheron, Enrique Gomez-Gomez, Alessia Cimadamore, Andreia Bilé-Silva, Rajan Gogna, Rodolfo Montironi, Liang Cheng

**Affiliations:** 1Department of Urology, Maimonides Biomedical Research Institute of Cordoba, University Hospital of Reina Sofia, UCO, 14004 Cordoba, Spain; 2Department of Morphological Sciences, University of Cordoba Medical School, 14004 Cordoba, Spain; 3Department of Medical Area (DAME), Institute of Pathological Anatomy, University of Udine, 33100 Udine, Italy; 4Urology Department, Egas Moniz Hospital, Centro Hospitalar de Lisboa Occidental, 1349-019 Lisbon, Portugal; 5Department of Human & Molecular Genetics, VCU Institute of Molecular Medicine (VIMM), VCU Massey Cancer Center, Virginia Commonwealth University School of Medicine, Richmond, VA 23298, USA; 6BRIC-Biotech Research & Innovation Centre, Faculty of Health and Medical Sciences, University of Copenhagen, 1165 Copenhagen, Denmark; 7Champalimaud Centre for the Unknown, 1400-038 Lisbon, Portugal; 8Molecular Medicine and Cell Therapy Foundation, Polytechnic University of Marche, 60121 Ancona, Italy; 9Department of Pathology and Laboratory Medicine, Brown University Warren Alpert Medical School, Lifespan Academic Medical Center, and the Legorreta Cancer Center at Brown University, Providence, RI 02903, USA

**Keywords:** bladder cancer, prognosis, risk categories, molecular subtypes, TP53, CCND1, MKI67, gene expression, NanoString

## Abstract

**Simple Summary:**

Our study used NanoString technology, a high-throughput platform measuring gene expression at the mRNA level to identify a set of genes predictive of clinical outcomes in bladder cancer patients. Twenty-seven differentially expressed genes were correlated with clinicopathological variables including molecular subtypes (luminal, basal, null/double-negative), histological subtypes (conventional urothelial carcinoma or carcinoma with variant histology), clinical subtype (NMIBC and MIBC), tumor stage category (Ta, T1 and T2-4), tumor grade, PD-L1 expression (high vs. low expression), and clinical risk categories (low, intermediate, high, and very high). Then, two risk models integrating the molecular subtypes and the level of expression of TP53, CCND1 and MKI67 were developed. These models provided a score ranging from 0 (best prognosis) to 7 (worst prognosis) that could be used to predict patient’ outcome and guide treatment decisions in bladder cancer.

**Abstract:**

This study evaluated a panel including the molecular taxonomy subtype and the expression of 27 genes as a diagnostic tool to stratify bladder cancer patients at risk of aggressive behavior, using a well-characterized series of non-muscle invasive bladder cancer (NMIBC) as well as muscle-invasive bladder cancer (MIBC). The study was conducted using the novel NanoString nCounter gene expression analysis. This technology allowed us to identify the molecular subtype and to analyze the gene expression of 27 bladder-cancer-related genes selected through a recent literature search. The differential gene expression was correlated with clinicopathological variables, such as the molecular subtypes (luminal, basal, null/double negative), histological subtype (conventional urothelial carcinoma, or carcinoma with variant histology), clinical subtype (NMIBC and MIBC), tumor stage category (Ta, T1, and T2–4), tumor grade, PD-L1 expression (high vs. low expression), and clinical risk categories (low, intermediate, high and very high). The multivariate analysis of the 19 genes significant for cancer-specific survival in our cohort study series identified TP53 (*p* = 0.0001), CCND1 (*p* = 0.0001), MKI67 (*p* < 0.0001), and molecular subtype (*p* = 0.005) as independent predictors. A scoring system based on the molecular subtype and the gene expression signature of TP53, CCND1, or MKI67 was used for risk assessment. A score ranging from 0 (best prognosis) to 7 (worst prognosis) was obtained and used to stratify our patients into two (low [score 0–2] vs. high [score 3–7], model A) or three (low [score 0–2] vs. intermediate [score 3–4] vs. high [score 5–7], model B) risk categories with different survival characteristics. Mean cancer-specific survival was longer (122 + 2.7 months) in low-risk than intermediate-risk (79.4 + 9.4 months) or high-risk (6.2 + 0.9 months) categories (*p* < 0.0001; model A); and was longer (122 + 2.7 months) in low-risk than high-risk (58 + 8.3 months) (*p* < 0.0001; model B). In conclusion, the molecular risk assessment model, as reported here, might be used better to select the appropriate management for patients with bladder cancer.

## 1. Introduction

Bladder cancer is a highly prevalent disease worldwide and an important public health problem [1,2]. Conventional urothelial carcinoma is the most common diagnostic category among bladder cancer patients. In clinical practice, urothelial carcinoma is classified as non-muscle-invasive bladder cancer (NMIBC) characterized by highly recurrent tumors with low progression capacity and, therefore, a high survival rate after standardized therapy [3]. This category includes urothelial carcinoma in situ and low-to-high-grade Ta and T1 cases, representing about 70% of urothelial carcinomas in the urinary bladder. Muscle-invasive bladder cancer (MIBC) typically includes T2–T4 disease. Adjuvant intravesical BCG immunotherapy is the standard of care in high-risk NMIBC following transurethral resection [4]. Neoadjuvant chemotherapy followed by radical cystectomy is the treatment of choice in MIBC, which may additionally require biomarker-guided immune checkpoint inhibitors (ICI), targeted therapies, or other novel drugs conjugates when locally advanced or metastatic [5,6,7].

At the molecular level, TCGA-derived data identify deregulation of the cell cycle, histone pathway, PI3K/AKT/mTOR pathway, and chromatin remodeling, the four major signaling pathways altered in 93%, 89%, 72%, and 64% of bladder urothelial tumors, respectively [8]. Luminal and basal molecular subtypes also emerged from early TCGA studies and served as the bases for subsequently reported taxonomic studies of bladder cancer [8]. Furthermore, the deregulation of the major signaling pathways and the molecular subtypes show implications in bladder cancer therapy and variable impact on the prognostic stratification of patients [9,10,11,12,13,14,15]. In addition to these mostly transcriptomic-derived classifications, a novel molecular classification of bladder cancer with prognostic and therapeutic implications, including luminal/basal/double negative (null) categories, recently emerged but using NanoString nCounter gene expression analysis instead [16].

Several studies have addressed the clinical application of limited gene-signature panels as prognostic/predictive biomarkers of response to therapy, showing a good correlation in the reported data [17,18,19].

NanoString nCounter gene expression analysis has been a less common subject of reports on using limited gene panels as prognostic/predictive biomarkers in TCGA bladder cancer. The study BASE47 compared the expression of the genes in high-grade urothelial carcinoma using RNASeq and NanoString technologies, and also validated a classifier for luminal and basal molecular subtypes based on NanoString and nCounter analysis in an independent dataset. The results of the study indicate that the classifier is effective in distinguishing between luminal and basal molecular subtypes [20]. The successful application of the Prosigna test, a NanoString-based classifier in the management of breast cancer patients, provides a rationale for the clinical application of molecular subtyping in urothelial carcinoma [21]. Two additional studies used NanoString gene expression analysis as a prognostic tool in patients with bladder cancer [22,23]. We aimed to generate molecular data providing clinically meaningful datasets to identify genes of potential clinical relevance in bladder cancer using RNA expression profiles and the nCounter (NanoString Technologies, Inc., Seattle, DC, USA) technology. The current study generated a 27-gene classifier to explore differences in terms of molecular, histological, and clinical bladder cancer subtypes, clinical risk categories, T-stage categories, pathologic grade, PD-L1 expression, and cancer-specific survival in a cohort series of 107 urothelial carcinomas of the bladder.

## 2. Material and Methods

### 2.1. Study Population

The hospital’s ethics committee approved this project (Act #274-ref 3800/2018), and signed informed consent was obtained for all patients and was conducted according to the principles outlined in the Declaration of Helsinki. The study analyzed data from a retrospective cohort of 107 fresh-frozen carcinoma samples from patients diagnosed with MIBC or NMIBC who had undergone transurethral bladder resection (TURB) or radical cystectomy at Reina Sofia University Hospital (Cordoba, Spain) between 2005 and 2014. Only patients with primary diagnosis receiving BCG with maintenance and/or pre-cystectomy neoadjuvant chemotherapy (following the therapeutic protocol applied at that time) were allowed. Following surgery, the selected samples were divided into two halves; one was snap-frozen and stored at −80 °C until processing, and the second one, formalin-fixed and paraffin-embedded, served for the assessment of histopathological variables and to establish an overall quality of the sample. All urothelial carcinoma samples were re-assessed, and eleven patients were excluded from the final cohort due to poor quality of RNA and/or limited tumor volume. Finally, 91 samples were selected for the current study. Additionally, specimens of adjacent normal bladder tissue from 5 patients, resected about 10 cm from the tumor lesion, were used as control.

All samples included in this study were subject to evaluation to confirm histopathologic variables by a dedicated uropathologist (ALB). Tumors were re-assessed following the 2022 WHO (World Health Organization, Geneva, Switzerland) classification of urologic tumors and the 8th edition of the AJCC (American Joint Committee for Cancer) [24,25]. The molecular subtype (luminal, basal, double negative) [16], histological subtype (variant histology), clinical subtype, T-category, tumor grade, PD-L1 expression, and clinical risk category [26] were included as clinicopathological variables. For this study, cancer-specific survival (CSS) was defined as the time from surgery to death caused by bladder carcinoma. Survival time was defined as the period between diagnosis and death. The patient’s follow-up was defined as the number of months from the surgical procedure to the date of the latest cystoscopy (or the last visit or death). Recurrence event was defined as the reappearance of a tumor after the initial treatment with at least one tumor-free cystoscopy interval. Progression event was defined as a shift to any higher stage (T1-T2-T3) in recurrent tumors or the appearance of metastases.

### 2.2. Sample Collection and RNA Extraction

Tissue fragments were obtained during routine surgical procedures and stored at −80 °C until processing. Total RNA was extracted from pulverized bladder tumor tissue using miRNeasy Mini Kit (Qiagen Inc., Valencia, CA, USA) according to the manufacturer’s protocol. Total RNA concentration was quantified using a Nanodrop ND-1000 spectrophotometer (NanoDrop Technologies, Inc., Wilmington, DE, USA). RNA quality was measured by the RNA integrity number (RIN) and the percentage of RNA fragments.

### 2.3. Gene Expression Custom Panel and NanoString Analysis

The mRNA gene expression was conducted using a customized NanoString’s nCounter^®^ Tag Set panel of 27 genes known to be altered in bladder cancer that is involved in different cellular processes such as the regulation of transferase activity, G1/S cell cycle progression, protein kinase activity, programmed cell death and senescence. NanoString analysis was performed according to the manufacturer’s instructions. Transcripts were counted using the automated NanoString nCounter system (NanoString Technologies, Seattle, WA). Counts were normalized using the nSolver Analysis Software (version 4.0) with the Advanced Analysis (module 2.0.115) plugin. The normalization was achieved by using internal positive and negative control probes and housekeeping genes. Internal positive and negative control probes were designed to detect the presence or absence of specific transcripts in the sample. The housekeeping genes (TBP, TUBA1B, ALAS1, ACTB, and SDHA) are genes that are commonly expressed across a wide range of cell types and conditions and were used as a reference for normalizing gene expression data. In this study, the normalization process involved background thresholding with a threshold count value of 20, which means that any counts below 20 were considered to be background noise and excluded from the normalization.

### 2.4. PD-L1 mRNA Quantification by RT-qPCR

PD-L1 and the housekeeping gene RPS23 (ribosomal protein S23) expression was performed using SYBR green quantitative RT-PCR with samples analyzed in duplicate and 40 cycles of amplification. The cycle quantification threshold (Ct) values of PD-L1 and RPS23 were estimated as the mean of the two measurements. Ct values were normalized by subtracting its value from the housekeeping gene RPS23 and from the Ct value of the target gene (∆Ct). Expression results were then reported as 40-∆Cq.

### 2.5. Statistical Analysis

All statistical analyses were performed with SPSS 25.0 (SPSS Inc., Chicago, IL, USA) and MedCalc Statistical Software version 17.6 (MedCalc Software bvba, Ostend, Belgium). Patient and clinical characteristics were summarized as numbers and percentages. Normalized data were generated using the nSolver Analysis Software. The Metaboanalyst 5.0 was used to generate the heatmaps, which were mean-centered and divided by the SD of each variable (scaled Z-score). Hierarchical clustering of RNA expression was performed using Euclidean distances and the Ward algorithm. The t-test and analysis of variance (ANOVA) were applied to identify differentially expressed genes. When false discovery rate (FDR)-adjusted *p*-value < 0.05, the genes were regarded as statistical significance.

The differentially expressed genes were dichotomized using the median and the receiver operating characteristic curve to determine the best cutoff point that allowed optimal separation between high versus low expression with maximum combined sensitivity and specificity. The univariate and multivariate Cox proportional hazards regression model were used to investigate the CSS-related differential expression genes and clinicopathological variables. The coefficients of the variables with statistical significance in the multivariate model were used to determine a weight (score) for each variable to build categories. The Kaplan–Meier curve and the log-rank test for cancer-specific survival were used to compare the survival between risk groups in both models. A *p*-value < 0.05 was considered statistically significant. To avoid a bias due to non-statistically significant sampling, the current cohort series of 107 patients was chosen through a random and representative sampling method, which reflects the characteristics of the larger population, with the results analyzed in the context of the study objectives and the literature review.

## 3. Results

Table 1 presents the characteristics of the 96 patients in the study diagnosed with conventional urothelial carcinoma (67 cases [73.6%]) or with variant histology (24 cases [26.4%]). Eleven patients were female (11.5%). A median age of 73 (+SD, 10.48) years was seen in the current cohort series, which included patients with high grade (69.2%) and high T-stage category (60.5%) with high or very high risk (70.4%), mainly with NMIBC (72.5%). Recurrence and progression were observed in 54.5% and 10.6%, respectively, of NMIBC cases. On follow-up, 31.9% of the patients died of disease or were alive with disease (46 + 40.51 months; median + SD; range 2–125). Luminal was the most common (71.4%) molecular subtype in the current series. Low PD-L1 expression was observed in 60% of the cases. Table 2 includes the 27 genes selected for the current study, and Figure 1 illustrates the relationship between them.

In Figure 2, the heat map represents the level of gene expression according to clinicopathologic variables, including molecular subtypes (luminal, basal, null/double negative), histological subtypes (conventional urothelial carcinoma, or carcinoma with variant histology), clinical subtype (NMIBC and MIBC), tumor stage category (Ta, T1, and T2–4), tumor grade, PD-L1 expression (high vs. low expression), and clinical risk categories (low, intermediate, high and very high) of urothelial carcinoma in the current series. Similarly, differentially expressed genes in the context of molecular, histological, clinical subtypes, tumor-stage category, tumor grade, PD-L1 expression, and clinical risk categories are presented in Figure 3 using t-test or ANOVA for comparisons.

Table 3 identifies a signature of 19 differentially expressed genes significant for CSS using the *t*-test; the receiving operating characteristic allowed the development of the optimal helpful cut-off to perform uni- or multivariate analysis, which included the 19 differently expressed genes significant for CSS (see also Table 3), supplemented with clinicopathological variables, molecular subtypes, and PD-L1 expression. The multivariate analysis identified an independent predictive signature, including the molecular subtype (*p* = 0.005) and the expression of TP53 (*p* = 0.0001), CCND1 (*p* = 0.0001), or MKI67 (*p* < 0.0001) genes. The relative weight of each of the four independent predictors to establish a risk score for CSS in bladder cancer patients is presented in Table 4. Figure 4 illustrates the risk scores for CSS as Kaplan–Meir plots and stratified as two (low [score 0–2] vs. high [score 3–7], model A) or three (low [score 0–2] vs. intermediate [score 3–4] vs. high [score 5–7], model B) risk categories using the molecular subtype and the expression of three cell-cycle-related genes (CCND1, TP53, and MKI67). Table 5 illustrates the molecular risk models A and B separated as NMIBC and MIBC clinical categories. In both models, NMIBC and MIBC patients are mostly associated with low-to-intermediate risk or high risk, respectively. This is particularly visible for low-risk (score 0–2) in which 97.7% were NMIBC with prolonged CSS and high-risk (score 5–7) with very short CSS (model A) (Figure 4).

## 4. Discussion

Cell cycle, histone pathway, PI3K/AKT/mTOR, and chromatin remodeling are the four major signaling pathways identified to be deregulated in bladder cancer according to TCGA-derived data, with cell-cycle-related gene alterations the most common, present in over 90% of bladder carcinoma cases [8]. Cell-cycle-related alterations were also identified by earlier molecular and immunohistochemistry-based studies that reported TP53, p21, p27, Ki67, cyclin D1, and D3, among other alterations, and frequently correlated with prognostic features in bladder carcinoma [76,77,78,79]. Cell-cycle-related alterations have been reported in other cancers including prostate cancer. These alterations, which can be detected in circulating tumor cells or DNA, may be of potential clinical utility in the management of these patients [80].

These studies highlight the potential of cell-cycle genes as research targets for molecular diagnostic panels. Further, there is a need for studies focusing on developing molecular signature panels for clinical decision support, improving prognostic accuracy, and guiding novel therapies in patients with bladder urothelial carcinoma. These studies should also take into account other risk factors such as meat intake, a significant risk factor for chronic diseases including bladder cancer onset pathogenically related to the generation of heterocyclic amines and polycyclic aromatic hydrocarbons by high-temperature cooking [81].

Following this rationale, several studies have addressed the clinical application of limited gene-signature panels as prognostic/predictive biomarkers of response to therapy, showing, in general, a good correlation with prognostic/predictive parameters [17,18,19,20,22,23,27,82,83,84,85]. A three-gene signature prognostically related with recurrence-free (RXRA and FGFR3) or progression-free (RXRA) survival in NMIBC, with low FGFR3 expression associated with good response to BCG intravesical instillations (showing no-to-late recurrence) was recently reported [19]. While the relationship between FGFR3 expression and BCG response has been established, the underlying mechanisms are not fully understood [86]. The study reported by Le Goux et al. also showed that mutations in commonly altered genes in bladder cancer such as HRAS, FGFR3, PIK3CA, and TERT were unassociated with the prognosis of NMIBC or MIBC [19]. Exploratory studies using different cohort series, including TCGA data, have also addressed small-sized panels showing variable prognostic sensibility [18)]. Additionally, luminal and basal molecular subtypes also emerged from TCGA early studies and served as the bases for subsequently reporting the taxonomic studies of bladder cancer [8]. In addition, the deregulation of the major signaling pathways and the molecular subtypes show implications in bladder cancer therapy and variable impacts on prognostic stratification of the patients [8,9,10,12,13,16,46]. The molecular subtypes’ clinical impact remains investigational at present [10,13]. In addition to these mostly transcriptomic-derived taxonomic classifications, a novel molecular classification of bladder cancer, including luminal/basal/double negative categories, recently emerged using NanoString nCounter gene expression analysis, also with important prognostic and therapeutic implications [16]. Furthermore, a recent report concurrently compared the so-called BASE47 genes in high-grade urothelial carcinoma using RNASeq and NanoString, the classifier for luminal and basal molecular subtypes based on NanoString and nCounter [20]. This analysis was validated using an independent dataset. The training and validation datasets accurately classified 87% and 93% of samples, respectively [20]. These results support luminal and basal molecular subtypes as potentially relevant clinical categories when classified by NanoString methods, thus providing a rationale for clinical application [16,20]. Prosigna test, a NanoString-derived classifier currently in use to manage breast cancer patients, is an example of the applicability [21]. Two additional studies using NanoString gene-expression analysis as a prognostic tool for patients with bladder cancer have been recently published. A seven-gene signature was recently applied to 138 MIBC cases and provided luminal and basal molecular subtypes. The classifier showed a high concordance with immunohistochemistry-derived molecular subtype of over 96% and correlated with disease-specific survival on multivariate analysis [23]. However, another study using 193 MIBC cases found no association with disease-specific or recurrence-free survival using NanoString gene expression analysis and a 21-gene panel. The retrospective nature and the limited number of patients are behind this study’s lack of prognostic association [22]. Heterogeneous results are a common finding in molecular-based studies, an important limitation in translating molecular data into clinical practice. The reasons behind are poorly understood, and may include the use of different and diverse methodologies and cut-offs, differences in study design, tissue sample issues, and probably tumor heterogeneity among others.

Our study, which is in line with previous reports [8,10,13,16,20,22,23], included the analysis of 27 bladder-cancer-related genes classifier, with selected genes from the current literature known to be related to the prognosis or as a potential therapeutic target of bladder cancer [19,27,28,29,30,31,32,33,34,35,36,37,38,39,40,41,42,43,44,45,46,47,48,49,50,51,52,53,54,55,56,57,58,59,60,61,62,63,64,65,66,67,68,69,70,71,72,73,74,75,87]. The goal was to explore differences in terms of molecular, histological, and clinical bladder cancer subtypes, clinical risk categories, T-stage categories, pathologic grade, and PD-L1 expression that might be useful to construct a prognostic/predictive gene signature panel useful in the clinic, mostly related to cancer-specific survival. Similarly, as suggested by the recent literature [88], liquid biopsy can detect biomarkers associated with histological and clinical subtypes, enabling non-invasive diagnosis and monitoring of disease progression, and that ctDNA level correlate with tumor stage and grade, and more advanced disease associated with higher ctDNA levels. Additionally, it can also be used in assessing changes in PD-L1 expression levels during treatment with immune checkpoint inhibitors [88]. Although liquid biopsy is not within the scope of the current study, we hypothesized that it might be used in the near future to stablish risk categories in bladder cancer, just similar to what we have observed in the current study using NanoString technology and tumor tissue samples.

Like other studies using NanoString nCounter gene expression analysis [16,20,21,22,23], we identified a gene signature of 19 differentially expressed genes significant for cancer-specific survival using optimal cut-off identified by the receiving operating characteristic of the tumor. Further analysis performed using these 19 CSS-related genes using multivariate analysis identified and independent predictive signature that included the molecular subtype and the expression of the cell-cycle-related genes TP53, CCND1, and MKI67. The relative weight of each of these four independent predictors was used to construct a risk-associated prognostic signature, thus proving a score for cancer-specific survival in bladder cancer patients. The model was then applied as two or three risk categories with marked differences in cancer-specific survival. Prolonged survival (median 116.89 months) was observed within low-risk category (score 0–2). By contrary, the high-risk category (score 5–7) presented a dismal prognosis with a low survival rate (median 4.44 months). Intermediate risk, as expected, was in between, showing median survival of 5 years.

Our study further supports the feasibility of NanoString technology to provide a clinically useful decision tool to provide accurately developed gene panels as prognostic molecular biomarkers in bladder cancer. The additive value of the molecular subtypes in the context of cell-cycle gene alterations is of relevance and represents original data derived from the current study. Importantly, NanoString technology may provide lower costs compared to transcriptomic technology and fast turnaround time [20,23]. Despite other scientific merits, the use of NanoString technology may significantly reduce the cost of molecular testing allowing a quick introduction of molecular testing in the assessment of bladder cancer patients. Finally, limitations of the current study include its retrospective nature with single-center data only, and the relatively small sample size. Nonetheless, the long follow-up (median of 46 ± 40.51, 2–125 months) of our cases may add value to the current series.

## 5. Conclusions

The current study was conducted using the novel NanoString nCounter gene expression analysis, which allowed us to identify the molecular subtypes and to analyze the gene expression of 27 bladder cancer-related genes. The differential gene expression correlated with clinicopathological variables, such as the molecular subtypes (luminal, basal, null/double negative), histological subtype (conventional urothelial carcinoma, or carcinoma with variant histology), clinical subtype (NMIBC and MIBC), tumor stage category (Ta, T1, and T2–4), tumor grade, PD-L1 expression (high vs. low expression), and clinical risk categories (low, intermediate, high and very high). For risk assessment, a scoring system based on the molecular subtype and the gene expression signature of TP53, CCND1, or MKI67 was applied, which allowed stratifying our patients into low, intermediate, or high risk of aggressive behavior. If validated by well-conducted studies, the proposed three-gene panel plus molecular subtype risk assessment model might be used as a clinical decision tool to select the appropriate management for bladder cancer patients.

## Figures and Tables

**Figure 1 cancers-15-02149-f001:**
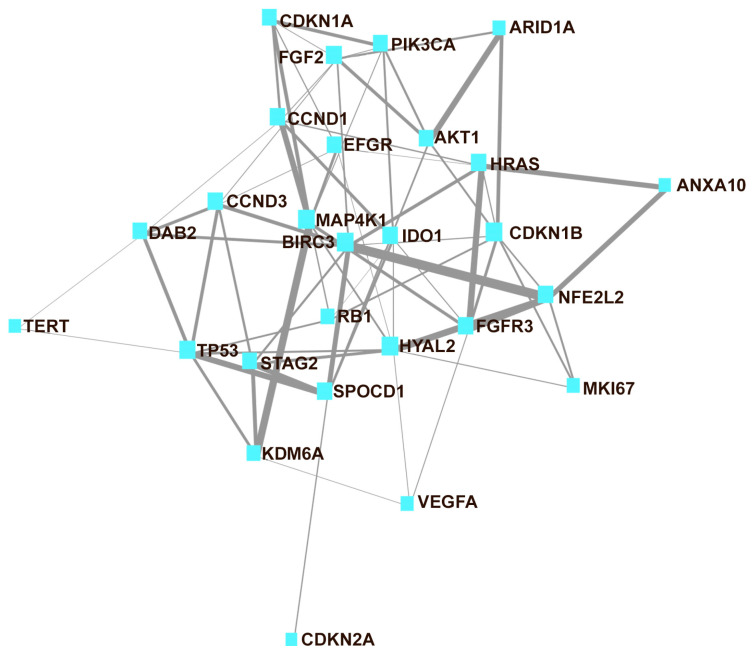
Network map of the 27 interconnected genes assayed in the current study. The thickness of the line is related to the strength of the association between genes (network analysis obtained using MetaboAnalyst version 5.0).

**Figure 2 cancers-15-02149-f002:**
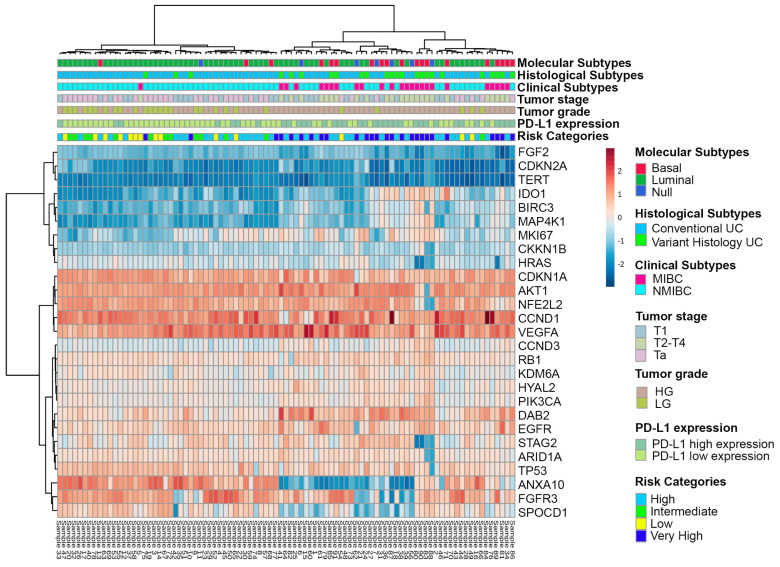
Heat map showing the gene expression level grouped as molecular, histological, and clinical subtypes, as tumor-stage category, tumor grade, PD-L1 expression, and clinical risk categories of urothelial carcinoma.

**Figure 3 cancers-15-02149-f003:**
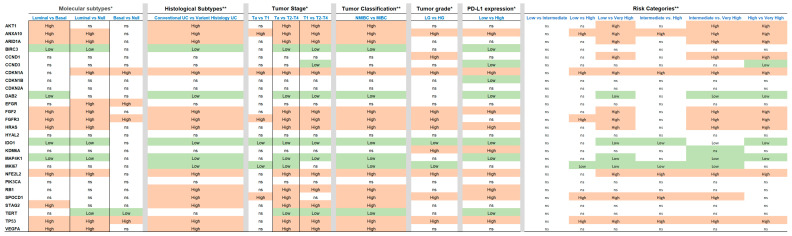
Statistical analysis based on *t*-test * or ANOVA ** showing associations of the genes differentially expressed with molecular, histological, and clinical subtypes of urothelial carcinoma and with tumor-stage category, tumor grade, PD-L1 expression or clinically meaningful risk categories (statistical significance based on false discovery rate *p* < 0.05).

**Figure 4 cancers-15-02149-f004:**
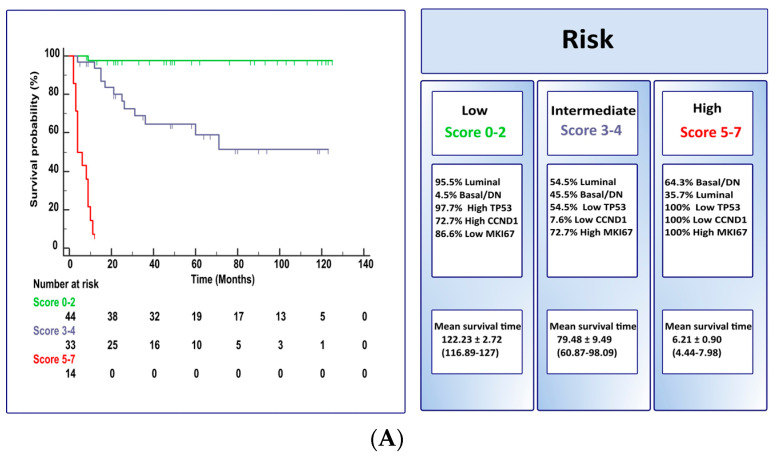
Kaplan-Meir plots for cancer-specific survival and risk scores stratified as two (*p* < 0.0001; model (**A**) or three (*p* < 0.0001; model (**B**) categories. The observed risk scores utilize the molecular subtype, the expression of two cell-cycle related genes (CCND1 and MKI67) and TP53 (See also Table 5 for details).

**Table 1 cancers-15-02149-t001:** Demography and clinicopathological characteristics of patients included in the study.

Variables	N (%)
**Control**	5 (5.2)
**Tumor**	91 (94.8)
**Gender**	
Male	85 (88.5)
Female	11 (11.5)
**Age, yr, median** **± SD (range)**	73 ± 10.48 (43–95)
**Followup (median ± SD, range), in months**	46 ± 40.51 (2–125)
**Molecular Subtypes**	
Luminal	65 (71.4)
Basal	19 (20.9)
Null	7 (7.7)
**Histologic Subtype**	
UC-conventional	67 (73.6)
UC-with variant histology	24 (26.4)
**Clinical Subtypes**	
NMIBC	66 (72.5)
MIBC	25 (27.5)
**Tumor stage ***	
Ta	36 (39.5)
T1	30 (33.0)
T2-T4	25 (27.5)
**Tumor Grade (WHO 2022)**	
High-grade	63 (69.2)
Low-grade	28 (30.8)
**PD-L1 expression**	
High expression	36 (40)
Low expression	54 (60)
**Risk categories ****	
Low	14 (15.4)
Intermediate	13 (14.3)
High	38 (41.8)
Very High	26 (28.6)
**Recurrence event in NMIBC**	
Yes	36 (54.5)
No	30 (45.5)
**Progression event in NMIBC**	
Yes	7 (10.6)
No	59 (89.4)
**Survival (NMIBC and MIBC)**	
NED	34 (37.4)
AWD	3 (3.3)
DBC	26 (28.6)
DOC	28 (30.7)

* Stage category based on AJCC/TNM 2016 revision. ** Risk stratification system based on WHO 2004 grading. NED: No evidence of disease. AWD: Alive with disease. DBC: Died of bladder cancer. DOC: Dead of other causes.

**Table 2 cancers-15-02149-t002:** Characteristics of the 27 bladder cancer-related genes in the study.

Gene Descriptor		References	
		Prognosis	Target Therapy
**AKT1**	AKT serine/threonine kinase 1	[27,28]	[29,30,31,32]
**ANXA10**	Annexin A10	[33,34]	
**ARID1A**	AT-Rich Interaction Domain 1A	[35,36,37,38,39]	
**BIRC3**	Baculoviral IAP Repeat Containing 3	[40,41]	
**CCND1**	Cyclin D1	[36,42]	
**CCND3**	Cyclin D3	[36,43]	
**CDKN1A**	Cyclin-dependent kinase inhibitor 1A (p21)	[27,44]	[44,45]
**CDKN1B**	Cyclin-dependent kinase inhibitor 1B (p27, Kip1)	[46,47]	
**CDKN2A**	Cyclin Dependent Kinase Inhibitor 2 (p16)	[27,38,48]	[48]
**DAB2**	Disabled homolog 2	[34,49]	[50]
**EGFR**	Epidermal growth factor receptor	[39,51,52]	[53]
**FGF2**	Fibroblast growth factor 2	[54,55]	
**FGFR3**	Fibroblast growth factor receptor 3	[19,27,36,37,39,56]	[53,57]
**HRAS**	HRas proto-oncogene, GTPase	[27,36,58]	[59]
**HYAL2**	Hyaluronidase 2	[34,60]	
**IDO1**	Indoleamine 2,3-dioxygenase 1	[61,62]	[62,63]
**KDM6A**	Lysine demethylase 6A	[27,56]	
**MAP4K1**	Mitogen-activated protein kinase kinase kinase kinase 1	[34,64]	
**MKI67**	Marker of proliferation Ki-67	[65,66]	
**NFE2L2**	NFE2 like bZIP transcription factor 2	[56,67]	
**PIK3CA**	Phosphatidylinositol-4,5-bisphosphate 3-kinase catalytic subunit alpha	[27,37,39,68]	[29,30,69]
**RB1**	RB transcriptional corepressor 1	[39,56]	
**SPOCD1**	SPOC domain containing 1	[34]	
**STAG2**	Stromal Antigen 2	[70,71]	
**TERT**	Telomerase reverse transcriptase	[27,72]	[73]
**TP53**	Tumor protein p53	[27,37,38,39,74]	[45,74]
**VEGFA**	Vascular endothelial growth factor A	[54,75]	[53]

**Table 3 cancers-15-02149-t003:** Cancer-specific survival-related gene expression signature.

Gene	*t*-Test	FDR	AUC	Optimal Cut-Off
HRAS	<0.0001	<0.0001	0.80	2.48
CDKN1A	<0.0001	<0.0001	0.80	3.4
MAP4K1	<0.0001	<0.0001	0.79	2.01
NFE2L2	<0.0001	<0.0001	0.79	3.43
TP53	<0.0001	<0.0001	0.84	3.06
FGF2	<0.0001	<0.0001	0.80	1.86
IDO1	<0.0001	0.0001	0.76	2.08
DAB2	<0.0001	0.0002	0.77	3.16
FGFR3	0.0001	0.0002	0.81	3.29
ANXA10	0.0003	0.0008	0.75	3.05
CCND1	0.0008	0.0020	0.76	3.77
AKT1	0.0011	0.0024	0.73	3.56
SPOCD1	0.0012	0.0025	0.74	2.87
ARID1A	0.0021	0.0041	0.68	2.93
MKI67	0.0023	0.0042	0.70	2.6
CCND3	0.0039	0.0065	0.71	2.71
BIRC3	0.0044	0.0070	0.68	2.28
VEGFA	0.0150	0.0225	0.66	3.72
STAG2	0.0173	0.0246	0.57	3.09

FDR: false discovery rate; AUC: Area under the ROC Curve; ROC Receiving operating characteristic.

**Table 4 cancers-15-02149-t004:** Weights are used to calculate the risk scores associated with cancer-specific survival. The analysis allows calculating the patient’s risk scores [range, 0 (best prognosis) to 7 (worst prognosis)] (see also Figure 4A,B for details).

Factors	β	HR	Score
**Molecular subtypes**	1.277	3.584	
Luminal			0
Basal/Null			1
**TP53 expression**	−2.408	0.090	
High			0
Low			2
**CCND1 expression**	−2.710	0.067	
High			0
Low			2
**MKI67 expression**	2.808	16.579	
High			2
Low			0
**Total score**			**0–7**

**Table 5 cancers-15-02149-t005:** Molecular risk models A and B presented as NMIBC and MIBC clinical categories (See also Figure 4).

**Model A**					
Low Risk (score 0–2)		Intermediate Risk (score 3–4)		High Risk (score 5–7)	
43 (97.7%) NMIBC		22 (66.7%) NMIBC		1 (7.1%) NMIBC	
1 (2.3%) MIBC		11 (33.3%) MIBC		13 (92.9%) MIBC	
**Model B**					
Low Risk (score 0–2)			High Risk (score 3–7)		
43 (97.7%) NMIBC			23 (48.9%) NMIBC		
1 (2.3%) MIBC			24 (51.1%) MIBC		

NMIBC: Non-muscle invasive bladder cancer. MIBC: Muscle-invasive bladder cancer.

## Data Availability

Data available on request due to privacy restrictions.

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
