# Peer review of "Risk Classification of Bladder Cancer by Gene Expression and Molecular Subtype"

_cancers, 2023, doi:10.3390/cancers15072149_

Round 1
Reviewer 1 Report
The study analyzes the expression of 27 genes in bladder cancer tissues, and how their expression is related to different histotypes, tumor stages and different risk categories. 4 genes were identified in a cohort of 107 patients, the expression of which appears to be an independent risk factor. The genes are TP53, CCND1 MKI67 and the molecular subtype (luminal or basal). They are used to establish a risk scale ranging from 0 to 7 (low - high risk). Furthermore, the overexpression or downregulation of some genes has been correlated with a better or worse prognosis, such as the FGFR3 gene whose downregulation is associated with a good response to BCG immunotherapy. If confirmed by well conducted studies, the molecular risk assessment model, might be used better to select patients with bladder cancer for the appropriate management.
My suggestion:
- English language should be improved in both grammar and syntax.
- The data were obtained on a cohort of 107 patients, so there could be a bias due to non-statistically significant sampling
- The article analyzes the role of the FGFR3 gene and the correlation between its downregulation and a good response to BCG therapy. At this regard i suggest this article: https://www.cancertreatmentreviews.com/article/S0305-7372(23)00021-X/fulltext
- The current study generated a 27-gene classifier to explore differences in terms of molecular, histological, and clinical bladder cancer subtypes, clinical risk categories, T-stage categories, pathologic grade, PD-L1 expression and cancer-specific survival. At this regard i suggest this article: http://pubmed.ncbi.nlm.nih.gov/34999017/
Cell cycle-related alterations were also identified by earlier molecular and immunohistochemistry-based studies that reported TP53, p21, p27, Ki67, cyclin D1, and D3, among other alterations, and frequently correlated with prognostic features in bladder carcinoma. These studies highlight the potential of cellcycle genes as research targets for molecular diagnostic panel At this regard i suggest this article: http://pubmed.ncbi.nlm.nih.gov/35805043/
Author Response
We would like to thank the reviewer for taking the time and effort to improve our manuscript.
- English language should be improved in both grammar and syntax.
English language grammar has been revised and edited accordingly through the text.
- The data were obtained on a cohort of 107 patients, so there could be a bias due to non-statistically significant sampling.
We agree with the reviewer in the fact that a small sample size may be associated with lacking statistical power. To clarify this issue, a new paragraph was added to the Statistical Analysis section, page#5, as follows:
…To avoid a bias due to non-statistically significant sampling, the current cohort series of 107 patients was chosen through a random and representative sampling method, which reflects the characteristics of a larger population, with the results analyzed in the context of the study objectives and the literature review.
- The article analyzes the role of the FGFR3 gene and the correlation between its downregulation and a good response to BCG therapy. At this regard i suggest this article: https://www.cancertreatmentreviews.com/article/S0305-7372(23)00021-X/fulltext
Following the reviewer suggestion, the new reference has been added on page# 12 and the references section (#86), as follows:
A 3-gene signature prognostically related with recurrence-free (RXRA and FGFR3) or progression-free (RXRA) survival in NMIBC, with low FGFR3 expression associated with good response to BCG intravesical instillations (showing no-to-late recurrence) was recently reported (19). While the relationship between FGFR3 expression and BCG response has been established, the underlying mechanisms are not fully understood (86). In the particular study reported by Le Goux et al. also showed that mutations in commonly altered genes in bladder cancer such as HRAS, FGFR3, PIK3CA, and TERT were unassociated with the prognosis of NMIBC or MIBC (19).
- The current study generated a 27-gene classifier to explore differences in terms of molecular, histological, and clinical bladder cancer subtypes, clinical risk categories, T-stage categories, pathologic grade, PD-L1 expression and cancer-specific survival. At this regard i suggest this article: http://pubmed.ncbi.nlm.nih.gov/34999017/
Following the reviewer suggestion, the new reference has been added on page# 13 and the references section (#90), as follows:
Similarly, as suggested by the recent literature (90), liquid biopsy can detect biomarkers associated with histological and clinical subtypes, enabling non-invasive diagnosis and monitoring of disease progression, and that ctDNA level correlate with tumor stage and grade, and more advanced disease associated with higher ctDNA levels. Besides, it can also be used in assessing changes in PD-L1 expression levels during treatment with immune checkpoint inhibitors (90). Although liquid biopsy is not within the scope of the current study, we hypothesized that might be used in near future to stablish risk categories in bladder cancer, just similar to what we have observed in the current study using NanoString technology and tumor tissue samples.
- Cell cycle-related alterations were also identified by earlier molecular and immunohistochemistry-based studies that reported TP53, p21, p27, Ki67, cyclin D1, and D3, among other alterations, and frequently correlated with prognostic features in bladder carcinoma. These studies highlight the potential of cellcycle genes as research targets for molecular diagnostic panel. At this regard i suggest this article: http://pubmed.ncbi.nlm.nih.gov/35805043/
Following the reviewer suggestion, the new reference has been added on page# 12 (bottom) and the references section (#80), as follows:
Cell cycle-related alterations have been reported in other cancers including prostate cancer. These alterations that can be detected in circulating tumor cells or DNA, may be of potential clinical utility in the management of these patients (80).
Reviewer 2 Report
The current study generated a 27-gene classifier to explore differences in terms of molecular, histological, and clinical bladder cancer subtypes, clinical risk categories, T-stage categories, pathologic grade, PD-L1 expression and cancer-specific survival in a cohort series of 107 urothelial carcinomas of the bladder. A revision is required.
- Adjuvant intravesical BCG immunotherapy is the standard of care in high-risk NMIBC following transurethral resection. A citation is required.
- I suggest shortening the introduction. Please restructure the introduction by shortening and eliminating data and comments from other studies that even if valuable should be included in the discussion. Indeed, some concepts are retracted in the discussion and this makes the whole text redundant and difficult to read.
- Risk classification could be also structured on modifiable risk factors. Among those, the meat intake seems to me impactful. Please discuss this interesting aspect including the following article: DOI: 10.3390/cancers14194775
- A syntax check required
- Check typos
Author Response
We would like to thank the reviewer for taking the time and effort to improve our manuscript.
- Adjuvant intravesical BCG immunotherapy is the standard of care in high-risk NMIBC following transurethral resection. A citation is required.
Following the reviewer suggestion, the new reference has been added on page# 2 and the references section (#4), as follows:
Adjuvant intravesical BCG immunotherapy is the standard of care in high-risk NMIBC following transurethral resection (4).
- I suggest shortening the introduction. Please restructure the introduction by shortening and eliminating data and comments from other studies that even if valuable should be included in the discussion. Indeed, some concepts are retracted in the discussion and this makes the whole text redundant and difficult to read.
Thank you very much for the comment. To avoid redundancy, we have deleted 3 paragraphs and reworded an additional one, thus resulting in a shortened version of the introduction.
- Risk classification could be also structured on modifiable risk factors. Among those, the meat intake seems to me impactful. Please discuss this interesting aspect including the following article: DOI: 10.3390/cancers14194775
Following the reviewer suggestion, the new reference has been added on page#12 and the references section (#81), as follows:
These studies should also take into account other risk factors such as meat intake, a significant risk factor for chronic diseases including bladder cancer onset pathogenically related to the generation of heterocyclic amines and polycyclic aromatic hydrocarbons by high-temperature cooking (81).
- A syntax check required
The text has already been checked through.
- Check typos
The text has already been checked through.
Reviewer 3 Report
In this study, the authors have investigated the association between gene expression profiles and subtypes of bladder cancer patients. The manuscript reads well except for some minor issues that would be beneficial to address for the readers' attention.
Comments
1. You should consider rephrasing the conclusion statement of the abstract or removing the phrase "If confirmed by well-conducted studies ". In the abstract, you would consider a brief outcome of the study, yet you can use it in the discussion section.
2. The term "frequent" doesn't read perfect for the concept despite giving the meaning. Please consider using a synonym for it.
3. Please revise the statement for completeness and clarity. Also, you can support the claim with references.
4. On page 3, you highlighted two opposite outcomes derived from the studies. Could you please discuss the potential reasoning for heterogeneity? In the recent form, the thought is not complete and therefore may cause to loss of attention to the reader.
5. In the hypothesis statement, the phrase "apply" doesn't fit well to the concept. Please consider using another verb to reflect your aim. Also, please extend the aim of the study in the first sentence.
6. You can release the name of the institution for IRB.
7. Could you please clarify the normalization method and its purpose regarding transcript counts?
8. The statement "When adjusted P‑value <0.05, the genes were regarded as statistical significance." needs to be revised for clarity.
9. The phrase "(Closest to the top-left corner)" can be removed without changing the meaning. ROC curves are widely used and the behavior is well-known by the researchers.
10. The statement is replicated "A p-value < 0.05 was considered statistically significant. " In general, you cannot use two different p-values to describe the significance.
11. Please clearly generate the limitations of the current study. You superficially mentioned them but not clearly.
Author Response
We would like to thank the reviewer for taking the time and effort to improve our manuscript.
- You should consider rephrasing the conclusion statement of the abstract or removing the phrase "If confirmed by well-conducted studies ". In the abstract, you would consider a brief outcome of the study, yet you can use it in the discussion section.
Page#2: The phrase "If confirmed by well-conducted studies” has been deleted from the abstract.
Page#2: A short paragraph on outcome added, as follows: Mean cancer-specific survival was longer (122+2,7 months) in low-risk than intermediate-risk (79.4+9.4 months) or high-risk (6.2+0.9 months) categories (p<0.0001; model A); and was longer (122+2.7 months) in low-risk than high-risk (58+8.3 months) (p<0.0001; model B).
- The term "frequent" doesn't read perfect for the concept despite giving the meaning. Please consider using a synonym for it.
The term “frequent” has been changed to “common” in the text.
- Please revise the statement for completeness and clarity. Also, you can support thchae claim with references.
Done
- On page 3, you highlighted two opposite outcomes derived from the studies. Could you please discuss the potential reasoning for heterogeneity? In the recent form, the thought is not complete and therefore may cause to loss of attention to the reader.
Thank you for the comment. The paragraph on page#3 has been deleted as suggested by reviewer#2 who recommended shorten the introduction section to avoid redundancy with respect to discussion section.
To clarify this issue, we have added a short paragraph on page 13, discussion section, as follows:
Heterogeneous results are a common finding in molecular based studies, an important limitation in translating molecular data into clinical practice. The reasons behind are poorly understood, and may include the use of different and diverse methodologies and cut-offs, differences in study design, tissue sample issues, and probably tumor heterogeneity among others.
- In the hypothesis statement, the phrase "apply" doesn't fit well to the concept. Please consider using another verb to reflect your aim. Also, please extend the aim of the study in the first sentence.
The term “apply” has been removed.
Page#3, bottom, reworded to extend the aim of the study in the first sentence, as follows:
We aimed to generate molecular data providing clinically meaningful datasets to identify genes of potential clinical relevance in bladder cancer using RNA expression profiles and the nCounter (NanoString Technologies, Inc., Seattle, DC, USA) technology. The current study generated a 27-gene classifier to explore differences in terms of molecular, histological, and clinical bladder cancer subtypes, clinical risk categories, T-stage categories, pathologic grade, PD-L1 expression, and cancer-specific survival in a cohort series of 107 urothelial carcinomas of the bladder.
- You can release the name of the institution for IRB.
The study was approved by the Local Ethical Committee (Act #274-ref 3800/2018) (University Hospital of Reina Sofia, Cordoba, Spain.). Also quoted in page#15
- Could you please clarify the normalization method and its purpose regarding transcript counts?
To clarify the normalization method and its purpose, a new paragraph has been added on page# 4, as follows:
The normalization was achieved by using internal positive and negative control probes and housekeeping genes. Internal positive and negative control probes were designed to detect the presence or absence of specific transcripts in the sample. The housekeeping genes (TBP, TUBA1B, ALAS1, ACTB, and SDHA) are genes that are commonly expressed across a wide range of cell types and conditions and were used as a reference for normalizing gene expression data. In this study, the normalization process involved background thresholding with a threshold count value of 20, which means that any counts below 20 were considered to be background noise and excluded from the normalization.
- The statement "When adjusted P‑value <0.05, the genes were regarded as statistical significance." needs to be revised for clarity.
The sentence has been modified for clarity on Page#5, as follows:
“When false discovery rate (FDR)-adjusted p-value <0.05, the genes were regarded as statistical significance”
- The phrase "(Closest to the top-left corner)" can be removed without changing the meaning. ROC curves are widely used and the behavior is well-known by the researchers.
The phrase “Closest to the top-left corner” has been removed from the text (page#5) and table 3.
- The statement is replicated "A p-value < 0.05 was considered statistically significant. " In general, you cannot use two different p-values to describe the significance.
The first p-value makes reference to FDR false discovery rate (FDR)-adjusted p-value <0.05 used to determine the statistical significance expression of genes, and the second p-value <0.05 is used to determine univariate and multivariate analysis.
- Please clearly generate the limitations of the current study. You superficially mentioned them but not clearly.
Final paragraph, page#14 reworded for clarity, as follows:
Finally, limitations of the current study include the retrospective nature with single-center data only, and the relatively small sample size. Nonetheless, the long follow-up (median of 46 ± 40.51, 2–125 months) of our cases may add value to the current series.
Reviewer 4 Report
The reviewer found the paper interesting and possibly clinically useful. A few specific questions/comments.
1. Given your identification of the three specific genes to follow as well as the differentiation of basal and luminal cancers do you feel that the basic approach could be used with gene arrays which might offer only relative quantification of genes. Or do you think that a quantitative approach requires nano string or RNA technologies. That is a separate question from whether nano string technology may have a shorter turn around.
2. Given your data do you think one could go back to published array samples which would include clinical outcome and confirm your present results.
3. The samples particularly as regards muscle invasive cancer appear slightly skewed . While Basal breast cancers represent 20% of your total cancers they represent a much higher percentage of your muscle invasive cancers . One would expect that but it brings up the question in Figure 3 are all the cases with scores of 5,6,and 7 muscle invasive. Then in the case of samples with scores of 0-2 are they all or virtually all non invasive? In certain regards the most significant group may be any non invasive lesions which meet your criteria to get a score of 3-4 since they may warrant the closest follow up and warrant therapies which may have more side effects.
Some minor discussion of certain of hese points would appear to be potentially useful.
Author Response
We would like to thank the reviewer for taking the time and effort to improve our manuscript.
- Given your identification of the three specific genes to follow as well as the differentiation of basal and luminal cancers do you feel that the basic approach could be used with gene arrays which might offer only relative quantification of genes. Or do you think that a quantitative approach requires nano string or RNA technologies. That is a separate question from whether nano string technology may have a shorter turn around.
Yes, we assume that the basic approach of identifying specific genes and differentiating between basal and luminal cancers can be applied using gene arrays, although comparison studies are not available. In fact, this would be a nice option to compare classic studies with novel technologies and to increase our knowledge on gene expression analysis in bladder cancer. Gene arrays are a cost-effective technology for analyzing gene expression levels, and they have been widely used in cancer research. On the other hand, using a quantitative approach may require more advanced technologies such as NanoString or RNA sequencing, not always available. These technologies provide more precise and accurate quantification of gene expression levels, which can be particularly useful for identifying subtle changes in gene expression that may be missed by gene arrays.
In terms of turnaround time, NanoString technology has been reported to have a shorter turnaround time compared to RNA sequencing, which can take several weeks to complete. However, the choice of technology ultimately depends on the specific research question to be answered and the experimental design, as well as factors such as cost, throughput, and level of sensitivity required.
- Given your data do you think one could go back to published array samples which would include clinical outcome and confirm your present results.
This is an interesting question and probably an open question. In our study, we have identified TP53, MIKi67 and CCND1, and basal/luminal as relevant genes in terms of expression. We and others have conducted studies on biomarkers at immunohistochemical level showing similar results, so if similar results are obtained with immunohistochemistry and NanoString, then it would be expected a confirmation of cell-cycle related genes using other technologies such gene arrays. This is just a hypothesis that needs to be substantiated but a priori makes sense.
- The samples particularly as regards muscle invasive cancer appear slightly skewed . While Basal breast cancers represent 20% of your total cancers they represent a much higher percentage of your muscle invasive cancers . One would expect that but it brings up the question in Figure 3 are all the cases with scores of 5,6,and 7 muscle invasive. Then in the case of samples with scores of 0-2 are they all or virtually all non invasive? In certain regards the most significant group may be any non invasive lesions which meet your criteria to get a score of 3-4 since they may warrant the closest follow up and warrant therapies which may have more side effects.
Thank you for the comments. To clarify this issue, we have prepared a new table and short paragraph, as follows:
End of page# 9 and beginning of page#10: Table 5 illustrates the molecular risk models A and B separated as NMIBC and MIBC clinical categories. In both models, NMIBC and MIBC patients are mostly associated with low-to-intermediate risk or high risk, respectively. This is particularly visible for low-risk (score 0-2) in which 97.7% were NMIBC with prolonged CSS and high-risk (score 5-7) with very short CSS (model A) (Figure 4).
Page#11: Table 5.
Table 5. Molecular risk models A and B presented as NMIBC and MIBC clinical categories (See also figure 4)
|
Model A |
|||
|
Low Risk (score 0-2) |
Intermediate Risk (score 3-4) |
High Risk (score 5-7) |
|
|
43 (97.7%) NMIBC |
22 (66.7%) NMIBC |
1 (7.1%) NMIBC |
|
|
1 (2.3%) MIBC |
11 (33.3%) MIBC |
13 (92.9%) MIBC |
|
|
Model B |
|||
|
Low Risk (score 0-2) |
High Risk (score 3-7) |
||
|
43 (97.7%) NMIBC |
23 (48.9%) NMIBC |
||
|
1 (2.3%) MIBC |
24 (51.1%) MIBC |
||
NMIBC: Non-muscle invasive bladder cancer
MIBC: Muscle-invasive bladder cancer
Round 2
Reviewer 1 Report
Authors answered all comments and suggestions
Reviewer 2 Report
Thank you for providing ad improved revised version. The manuscript is suitable for publication